# Sulphoraphane Affinity-Based Chromatography for the Purification of Myrosinase from *Lepidium sativum* Seeds

**DOI:** 10.3390/biom12030406

**Published:** 2022-03-05

**Authors:** Helena Galádová, Zoltán Polozsányi, Albert Breier, Martin Šimkovič

**Affiliations:** 1Faculty of Chemical and Food Technology, Institute of Biochemistry and Microbiology, Slovak University of Technology, Radlinského 9, 812 37 Bratislava, Slovakia; helena.galadova@stuba.sk (H.G.); zoltan.polozsanyi@stuba.sk (Z.P.); albert.breier@stuba.sk (A.B.); 2Centre of Biosciences, Institute of Molecular Physiology and Genetics, Slovak Academy of Sciences, Dúbravská cesta 9, 845 05 Bratislava, Slovakia

**Keywords:** myrosinase, *Lepidium sativum*, purification, affinity chromatography, sulforaphane as ligand, enzymatic properties

## Abstract

Sulforaphane and other natural isothiocyanates released from the respective plant glucosinolates by the plant enzyme myrosinase (β-thioglucoside glucohydrolase) show extensive anticancer and antimicrobial effects. In this study, myrosinase from garden cress (*Lepidium sativum*) seeds was purified to electrophoretic homogeneity by a fast and easy strategy consisting of fractionation by isoelectric precipitation with ammonium sulphate (AS) and affinity chromatography using sulforaphane (SFN) attached to cellulose resin. The overall purification of enzyme with respect to crude extract was 169-fold and recovery of 37%. Under non-reducing conditions, two protein bands exhibiting myrosinase activity with masses of about 114 and 122 kDa, respectively, and a 58 kDa protein band with no activity were detected by SDS-PAGE and zymography on polyacrylamide gel. MALDI-Tof/Tof of tryptic fragments obtained from the respective protein bands detected sequence motifs homologous to the regions responsible for glycoside-substrate binding and similarities to members of the enzyme subfamilies β-glucosidases and myrosinases GH. The enzyme hydrolyzed both the natural (sinigrin, sinalbin, glucoraphanin) and the synthetic (p-nitrophenol-β-D-glucopyranoside (pNPG)) substrates. The highest catalytic activity of purified enzyme was achieved against sinigrin. The K_M_ and V_max_ values of the enzyme for sinigrin were found to be 0.57 mM, and 1.3 mM/s, respectively. The enzyme was strongly activated by 30 μM ascorbic acid. The optimum temperature and pH for enzyme was 50 °C and pH 6.0, respectively. The purified enzyme could be stored at 4 °C and slightly acidic pH for at least 45 days without a significant decrease in specific activity.

## 1. Introduction

Myrosinases (β-thioglucoside glucohydrolase, EC 3.2.1.147) are a family of enzymes with β-glucosidase activity that is involved in the cleavage of S-glycosyl and O-glycosyl bonds [1] and release toxic volatile compounds such as nitriles, epitionitriles, thiocyanates, isothiocyanates and others from various plant glucosinolates [2]. The final product is dependent on the chemical structure of the glucosinolate aglycone part, the presence of metal ions, and the reaction conditions and presence of different effectors including specific proteins. Under normal circumstances, myrosinase and glucosinolates are physically separated from each other and are stored either in different cell compartments [3] or different plant tissues [4] to avoid an undesirable hydrolysis. In a cruciferous plants, myrosinase and glucosinolates are a prominent part of the defense mechanism that is activated by the mechanical damage of plants caused by herbivores or insects [5]. In addition, glucosinolates and their hydrolytic products are involved in allelopathic interactions among plants, which are applied in the agriculture industry and ecological farming for crop protection [6,7].

Although the antimicrobial and anticancer effects of plant isothiocyanates have been known for more than 50 years [8], these substances have been subjected to intensive research in recent decades (reviewed in [9]). It has been found that aliphatic isothiocyanates such as SFN or allyl isothiocyanate (released by myrosinase reaction from glucoraphanin or sinigrin) may affect the expression and function of antioxidant and detoxification cellular processes through deregulation PI3K/AKT/mTRK, MAP and NF-κB regulatory pathways [10]. The antitumor effects of isothiocyanates result from multilevel modification of various biological regulatory pathways. Their efficacy may vary in different neoplastic cells and may induce arrest of cell cycling [11], induction of apoptosis [11], or autophagy [12], as well as many other processes that limit disease progression.

The activity of natural isothiocyanates can be deduced from the electrophilic nature of the –NCS group, which reacts efficiently with the nucleophilic groups –SH, –NH_2_ and –OH [13]. Therefore, the parent glucosinolates have low potency and must be activated to cleavage products by the myrosinase reaction (reviewed in: [14]), which appears to be a crucial point for achieving the beneficial effects of isothiocyanates [15]. As the content of myrosinase in stored plants decreases with storage time [16], it seems advantageous to enrich glucosinolate-rich plant nutrient products by the addition of myrosinase.

The use of the health-promoting benefits of SFN has led to increased interest in the purification and characterization of plant myrosinases. Most currently used strategies for purification of myrosinases from plant material are derived from its structural and molecular properties, such as high degree of glycosylation of myrosinases, association with various myrosinase binding proteins into multiprotein complexes often reaching molecular weights up to 1000 kDa, presence of polar and dissociable amino acid residues in the enzyme molecule. To achieve the highest enzyme purity, chromatographic methods with different separation principles are mutually combined. The widest used method for purification of myrosinase from *Brassicaceous* seeds is the lectin binding chromatography using the interaction of concanavalin A with oligosaccharides attached on the enzyme molecule by N-glycosylation [17,18]. Another frequently used approach for myrosinase purification, which makes it possible to separate the high-molecular complexes of myrosinase and myrosinase-binding proteins, is gel-permeation chromatography [17]. Similarly, ion-exchange chromatography has often been employed as a step during the purification of plant myrosinases, either for the capturing or polishing of enzymes [19,20]. However, several unusual procedures, such as aqueous two-phase counter-current chromatography (CCC) system, used for the purification of myrosinase from daikon sprouts (*Raphanus sativus*) and broccoli sprouts (*Brassica oleracea* var. *italica*) [21], or isoelectric focusing, used for the purification of myrosinase from *Sinapis alba* [22], have been applied too.

In this study, we report that germline myrosinase from *Lepidium sativum* can be efficiently purified on SFN (as an affinity ligand) immobilized on aminocellulose by reaction with an NCS group. The characterization of the enzymatic and molecular properties of the purified myrosinase was further evaluated.

## 2. Materials and Methods

Unless otherwise indicated, the chemicals were from Merck spol. s r.o., Bratislava, Slovakia.

### 2.1. Plant Material, Its Homogenization and Preparation the Myrosinase Fraction by Isoelectric Precipitation by AS

Myrosinase was isolated from garden cress (*L. sativum*) seeds purchased commercially (Garden Seeds BV, Enkhuizen, Netherlands). Seeds (10 g) were surface sterilized by the three-step wash procedure (5 min wash in 70% (*v/v*) ethanol, 10 min wash in 3.125% NaOCl, and the final wash 3 times in excess of sterile tap water) and cultivated on the top of the wetted filter paper at room temperature (24 °C) for 24 h, under aseptic conditions. Swollen seeds were collected and suspended in 500 mL of the homogenization medium (20 mM Tris-HCl buffer, pH 7.4, 1 mM Na_2_ATP, 1 mM DTT, 1% (*v/v*) glycerol, 0.05% (*v/v*) Triton X-100, 1 mM PMSF, 1 mM benzamidine, and 1 μM pepstatin A) precooled on ice. After mixing seed suspension with 30 g of glass beads (with diameter of 0.1 to 0.3 mm), the suspension was disintegrated by grinder (ETA Sapelo 5013 90010, Prague, Czech Republic) in 10 cycles (30 s homogenization phase alternated by 60 s phase of chilling in an ice-water mixture). The homogenate was centrifuged at 50,000× *g* (Avanti J-30I, Beckman Coulter, Brea, CA, USA) for 30 min at 4 °C, and the resulting supernatant was used to prepare the protein fraction with myrosinase activity by precipitation with AS. 

First, pH of supernatant was adjusted by addition of MES-NaOH buffer, pH 6.8, to the final concentration of 150 mM and, then AS was gradually added to 45% (*w/v*) saturation. The suspension was allowed to gently stir at 4 °C for 12 h, and the precipitate was collected by centrifugation at 50,000× *g* as above. Protein pellet was dissolved in 200 mL of 20 mM Tris-HCl buffer, pH 7.4, containing 50 mM KCl, 5% (*v/v*) glycerol, 1 mM PMSF, 1 mM benzamidine and 1 μg mL^−1^ pepstatin A and dialyzed against 2 L of the same buffer at 4 °C with three changes every 4 h. After dialysis, the precipitated proteins were centrifuged down at 1300× *g* (Centrifuge 5430R, Eppendorf, Hamburg, Germany) for 10 min at 4 °C and supernatant were used for purification of myrosinase by affinity chromatography.

### 2.2. Purification of Myrosinase on Affinity Chromatography Column with SFN Ligand

The protein solution obtained after dialysis was applied to a cellulose column (1.0 × 5 cm) with SFN as ligand coupled through a diethyltriamine spacer via amino group (DETA + SFN) equilibrated at a flow rate of 0.5 mL/min with 12 column volumes (CV) of a mobile phase consisting of 20 mM Tris-HCl buffer, pH 7.4, 1 mM MgCl_2_, 1 mM DTT, 50 mM KCl (equilibration medium, EQM). After the removal of unbound proteins by washing with EQM, the column was eluted stepwise in three isocratic steps: in the first step with EQM containing 2 M KCl, followed with EQM containing 0.5% (*v/v*) Triton X-100 and in the final step with EQM containing 1 M guanidine thiocyanate. Washing in each step was carried out until the protein concentration and enzyme activity in the mobile phase did not change. Fractions eluted by Triton X-100 were pooled, the volume reduced 10-fold by ultrafiltration through a Biomax polyethersulfone membrane with a molecular weight cut-off of 30 kDa (Millipore, Bedford, MA, USA), and the detergent removed on a column of Bio-Beads SM-2 (Bio-Rad laboratories, Hercules, CA, USA) in EQM. The enzyme was concentrated by the Vivaspin centrifugal filter device (molecular weight cut-off of 30 kDa, Sartorius Stedim Biotech GmbH, Goettingen, Germany) to ~0.5 mg mL^−1^, and glycerol was added to 25% (*v/v*). The enzyme was stored at 4 °C. All purification steps were performed at 4 °C.

### 2.3. Preparation of Affinity Chromatography Column with SFN as Ligand

Cellulose beads (particle size of 80 to 100 μm) with covalently attached diethyl triamine spacers terminated by free amino groups (DETA) (Iontosorb, Usti nad Labem, Czech Republic) was used to prepare a SFN-specific affinity sorbent according to following reaction (Figure 1).

DETA matrix (3 g of wet matrix) was washed in 2 steps: first, with excess of deionized water; then, with 100 mM Na_2_CO_3_-NaHCO_3_ buffer, pH 9.0 (coupling medium), and suspended in the same buffer. Thereafter, DETA matrix suspension was mixed with 60 µL (0.4 mmol) of SFN and statically incubated for 16 h at 4 °C. After incubation, the DETA matrix with SFN (DETA + SFN) was washed with an excess of coupling medium, deaerated, and poured into a column (5 × 1 cm). The amount of SFN bound to DETA matrix was calculated on the basis of determination of the amount of unbound SFN using 1,2-benzenedithiol (1,2-BDT) [23,24]. 1,2-BDT reacts with SFN in a slightly alkaline medium to form 1,3-benzodithiol-2-thione, which is stable and absorbs in the UV region at 365 nm (ε_365_ = 23,000 M^−1^.cm^−1^).

### 2.4. Protein Concentration Determination

Proteins were routinely determined by the method of Bradford [25] with bovine serum albumin (BSA) as standard. Proteins in samples containing Triton X-100 were determined by the modified Lowry method, as described by Wang and Smith [26].

### 2.5. Assay for Myrosinase Activity and Characterization of Myrosinase Properties

Myrosinase activities were measured spectrophotometrically by the formation of glucose released from sinigrin hydrolysis. The amount of glucose was routinely assayed by the glucose oxidase-peroxidase coupled enzyme (GOP) method.

The reaction mixture (100 μL) contained 150 mM MES-NaOH buffer (pH 6.8), 100 µM L-ascorbic acid, and 0.2 mg mL^−1^ isolated proteins enriched on myrosinase after various steps of purification see above. The reaction was triggered by the addition of 5 mM sinigrin (as substrate) and, after stirring the mixture, conducted at 37 °C for 30 min. The measurement was stopped by boiling for 5 min in the water bath. When the zero-time point (at 0 min) was measured, the reaction mixture was boiled first and then substrate was added. The cooled reaction mixture was centrifuged at maximal velocity (12,000× *g*) (MiniSpin, Eppendorf, Hamburg, Germany) for 2 min at room temperature and supernatant taken for the determination of glucose concentration by the GOP method using a GLU1000 kit (Erba Lachema, Brno, Czech Republic). Supernatant, filled up with deionized water to a volume of 200 μL, was then mixed up with 800 μL of the GOP reagent and incubated at 37 °C for 15 min. After incubation, the absorbance of the mixture at 500 nm was read against the blank (the mixture of water with the GOP reagent). The concentration of D-glucose in the reaction mixture after 30 min and at zero-time point was estimated on the basis of calibration curve for glucose (expressed as the dependence of absorbance at 500 nm on the glucose concentration) obtained by the same procedure using standard solutions of glucose in the range from 0 to 200 μM. Myrosinase activity was defined as the amount of enzyme that catalyzed the release of 1 μmol of glucose from sinigrin per min under the assay conditions. Specific activity was expressed as the ratio of activity and the protein amount (in μmol/min/mg). Myrosinase assays were performed in triplicate, and the values shown represent the average and standard deviation.

To characterize the effect of pH on myrosinase activity, the following buffers (at 150 mM concentration in the reaction mixture) were used for pH adjustment: glycine-HCl over the pH range of 4.0 to 5.0, MES-NaOH for the range of pH 5.0 to 7.0, and Tris-HCl for pH of 7.0 to 9.0. The other assay conditions, as well as the procedure, were the same as described above. The effect of temperature on the myrosinase activity was measured under the same conditions as the routine assay, but at various temperatures ranging from 5 to 90 °C. When the thermal stability of enzyme was estimated, the reaction mixture was pre-incubated at the desired temperature (between 4 and 90 °C) for 60 min. After pre-incubation, the reaction mixture was cooled to 37 °C and the measurement of the residual myrosinase activity was started by sinigrin addition. All other conditions were the same as the routine enzyme activity assay mentioned above. The influence of ascorbate on the myrosinase activity was analyzed in the concentration range of 0.001 to 10 mM. When the kinetic parameters of the myrosinase towards glucosinolate substrates (sinigrin, glucoraphanin and sinalbin) were determined, the concentration of purified myrosinase was adjusted to 20 μg mL^−1^, the reaction time was lowered to 10 min, and the myrosinase activity was measured under conditions optimal for myrosinase activity (pH 6.0, 37 °C, ascorbate concentration of 100 μM). Up to 10 min, the time course of the glucose concentration changes was linear, making it possible to calculate the initial velocity of reaction. The kinetic parameters (V_max_ and K_M_) for myrosinase were determined by non-linear fitting the initial velocity data (dependence of initial velocity on substrate concentration) to the Michaelis–Menten equation in the Origin 8.5 program (Origin Lab Corporation, Northampton, MA, USA).

### 2.6. Electrophoresis of Proteins and Zymography Detection of Myrosinase Activity under Non-Denaturated Conditions

To accomplish the analysis of proteins and enzyme activities under non-denatured conditions, protein samples were analyzed by electrophoresis on 10% polyacrylamide gel (PAGE) at the presence of SDS by the Laemmli method [27] with three key modifications: protein-reducing agents (mercaptoethanol or dithiothreitol) were omitted from the sample buffer (1); the protein sample, after mixing with the sample buffer, was not subjected to thermal denaturation prior to loading into gel (2); and electrophoresis was run in the cold room at 4 °C with pre-chilled running buffer (3). After complete electrophoresis, the protein bands were stained with Coomassie Brilliant Blue R-250 dye according to standard procedure. At zymographic detection of myrosinase activity, polyacrylamide gel was washed (3 times for 20 min) with the excess of 20 mM MES-NaOH buffer, pH 6.0, containing 2.5% (*v/v*) Triton X-100 to remove SDS and renature enzyme activity. The detection of myrosinase activity was performed by incubation of the gel in buffered solution of 1 mM sinigrin in the presence of barium acetate according to the procedure of Bones and Slupphaug [28]. Zones with myrosinase activity were shown as subtle white precipitates of barium sulfate at the surface of gel.

### 2.7. Identification of Proteins by MALDI-MS

Identification of protein was performed according to the method of Shevchenko et al. [29]. After complete electrophoresis on the polyacrylamide gel, the protein band co-migrating with myrosinase activity was excised from the gel, trypsinized in-gel by sequencing grade modified porcine trypsin (Roche, Mannheim, Germany), and peptide fragments obtained were desalted on a C18 ZipTip column (Millipore, Bedford, MA, USA) according to the manufacturer’s instructions. Desalted peptide fragments, dissolved in sample medium (50% (*v/v*) acetonitrile (ACN) containing 0.1% (*v/v*) trifluoroacetic acid (TFA)) were mixed with a matrix solution (1:1 ratio, *v/v*) of α-cyano-4-hydroxycinnamic acid (CHCA) (10 mg/mL prepared in 50% (*v/v*) ACN and 0.1% (*v/v*) TFA) on a MALDI target plate [29]. Peptide mass spectrometry analysis was performed in the positive reflectance mode on AUTOFLEX III Smartbeam MALDI-Tof/Tof instrument (Bruker Daltonics, Bremen, Germany). The MS data were analyzed by the Mascot software (Matrix Science Ltd London, UK) using the NCBInr and/or Swiss-Prot databases for protein searching. The parameters for the Mascot search were set as follows: green plants as taxonomy, trypsin as the proteolytic enzyme with single missed cleavage, cysteine carbamidomethylation as fixed and methionine oxidation as variable modifications, peptide tolerance of 0.5 Da, fragment mass tolerance of 0.5 Da, and peptide charge of +1. A protein was considered to have been successfully identified if it had hit with a random match probability lower than 0.05. The most intense peaks from the observed mass spectrum of tryptic peptides were selected and subjected to fragmentation and the fragments analyzed by tandem mass spectrometry. The resulting MS/MS data were used for protein identification by the Mascot search.

The peptide sequences obtained by MS/MS analysis were compared to the amino acid sequences of plant myrosinases and β-D-glucosidases stored in NCBI database using the ClustalX program (available in: http://www.clustal.org/clustal2/ (accessed on 14 January 2022); for details see [30]).

### 2.8. Chemicals

ACN, SFN, 1,2-BDT, Tris, MES, DTT, benzamidine, pepstatin A, PMSF, glycerol, guanidine thiocyanate, sinigrin, glucoraphanin, sinalbin, p-nitrophenyl-β-D-glucopyranoside (pNGP/p-NP-β-D-Glc), p-nitrophenyl-α-D-glucopyranoside (p-NP-α-D-Glc), p-nitrophenyl-β-D-galactopyranoside (p-NP-β-D-Gal), o-nitrophenyl-β-D-galactopyranoside (o-NP-β-D-Gal), p-nitrophenyl-α-D-manopyranoside (p-NP-α-D-Man), acrylamide, N,N′-methylene-bisacrylamide, TEMED, ammonium persulfate, mercaptoethanol, SDS, glycine, L-ascorbic acid, Coomassie Brilliant Blue R-250, BSA, Triton X-100, TFA were purchased from Sigma-Aldrich (St. Louis, MO, USA). Na_2_ATP, Na_2_NADP^+^, hexokinase and glucose-6-phosphate dehydrogenase was from Roche Diagnostics (Mannheim, Germany). CHCA was from Bruker Daltonics (Bremen, Germany). All other chemicals were of analytical reagent purity grade and purchased from common commercial sources.

## 3. Results and Discussion

### 3.1. Purification of Myrosinase from L. sativum Seeds

Preliminary measurement of myrosinase activity in cell-free homogenates from swollen seeds of *Brassicaceae* family (results not shown) has shown that the seeds of garden cress (*L. sativum*) is a suitable source of myrosinase for purification. In an effort to prepare the myrosinase with purity as close as possible to electrophoretic homogeneity, the combination of two methods, isoelectric precipitation with AS (in the first step) and affinity chromatography (in the second step), were used for enzyme purification. At isoelectric precipitation, the two main parameters, the pH of precipitation medium and the saturation level of AS, were followed up. The purity and yield of myrosinase activity was measured in media with a pH range from 3.7 to 9.0 at 45% AS saturation. As seen in Figure 1A, the highest retention of myrosinase activity in the precipitate and its lowest content in the supernatant was observed at pH of 6.8.

At the same time, the proportion of protein in the precipitate obtained from the total protein content of the crude homogenate at pH 6.8 decreased to 57% (Figure 1A), which together with the increase in enzyme activity gives the myrosinase purification factor at 1.8. Precipitation carried out under more acidic (in the range of 3.7 to 6.1) or more alkaline conditions (in the range of 7.5 to 9.0) led to an increase in the portion of proteins in the pellet fractions (Figure 1A), and hence the degree of purity of the myrosinase fractions fell. These observations were also supported by zymographic analysis of the myrosinase activity of the protein bands in the electrophoretic gel in the precipitate fractions obtained at different pH values, which revealed slightly higher signals at pH 6.8 and 7.5 (Figure 1B). 

When the saturation level of AS in the precipitation medium was brought to 65%, the total myrosinase activity in the pellet fraction (at pH of 6.8) increased to 81.2% and the content of proteins was elevated to the similar extent. Although the specific myrosinase activity in the pellet fraction, obtained at the 65% level of AS, was comparable to one prepared at the 45% of AS, the SDS-PAGE electrophoresis revealed that the final myrosinase product obtained after the complete purification procedure of the proteins sedimented at 45% saturation contains less minor proteins (not shown). Therefore, the protein fraction precipitated at the 45% saturation of AS was used as the starting material for myrosinase purification by chromatography. 

The crucial step of the myrosinase purification from *L. sativum* seeds was one-step affinity chromatography using the macroporous beaded cellulose with chemically linked SFN as ligand obtained by formation of disubstituted thiourea from primary aminogroup of diethyltriamino side arm of DETA cellulose and NCS group of SFN (see Section 2). The reason for using SFN as ligand was the ability of methylsulfinylbutyl portion of SFN (aglycone part of glucoraphanin) to interact with non-polar amino acids residues of hydrophobic pocket of the active site of myrosinase [31,32,33].

The presence of immobilized ligand in the obtained cellulose derivative has been proved by the FTIR spectroscopy. The measured spectra of both types of matrix, DETA unmodified and DETA modified with SFN, showed several characteristic absorption peaks, at wavenumbers of 3340, 1424, 1371, 1166, and 896 cm^−1^ (Figure 2A), assigned to cellulose [34], the major constituent of the matrices. The dominant functional group of SFN is the sulfinyl group with the absorbance band at 1025 cm^−1^ [35,36]. Changes in the 1024 cm^−1^ region are observable in the absorption spectra of derivatized DETA compared to the original DETA (Figure 2A).

A significant absorption maximum at a wavelength of 1024 cm^−1^ is observed in the differential absorbance spectrum, which is the difference between the absorbance spectrum of the derivatized SFN-bound DETA and the original DETA (both obtained under the same conditions, Figure 2B). Other bands with wavenumbers, at 692, 1272, 1365, 1454 cm^−1^, typical for SFN [35,36] and, at 1090, 1417 cm^−1^, typical for thiourea link [37] were observed in the difference spectrum too, but with much lower intensities. The amount of bound SFN 0.3 mg per g dry DETA was determined from the material balance of total SFN added to the reaction with DETA and the amount of SFN remaining free in solution after the reaction. The concentration of free SFN was determined according to Zhang et al. [24].

After loading the protein fraction obtained at 45% saturation of AS onto the column filled with the DETA + SFN resin, the column was first washed by the equilibration medium (EQM) to elute unbound material and then sequentially washed by the EQM containing three different stripping agent, 2M KCl, 0.5% (*v/v*) Triton X-100 and 1M guanidine thiocyanate, particularly, to elute bound proteins (Figure 3A). 

A portion of the myrosinase activity did not bind to the column and flow through the column in the void volume, another portion was bound and could be eluted from the column during washing with 0.5% Triton X-100 (Figure 3). A small proportion of myrosinase activity also left the column during washing with 2 M KCl, which also indicated the proportion of ionic/electrostatic interactions in enzyme binding to derivatized DETA. The enzyme portion of interest was the one wholly found in the fraction bound to column and released with the non-ionic detergent Triton X-100. The presence of Triton X-100 in the final enzyme preparate, obtained after being concentrated, did neither affect the enzyme stability nor the assaying of enzyme activity, but interfered with the accurate estimation of protein concentration; hence, the detergent concentration decreased on a level that did not affect the protein measurement. It was observed that complete detergent removal from purified myrosinase lead to strong losing of the enzyme activity.

The typical recovery and purification factor of myrosinase from *L. sativum* seeds is presented in Table 1.

The protein portion recovered from precipitation by addition of 45% AS to the crude extract accounted for approximately half of the total enzyme activity (51.7%). This step improved myrosinase purity almost 2-fold. The AS fraction was further resolved by affinity chromatography using SFN as ligand into three peaks that exhibited myrosinase activity and the protein portion bound to derivatized DETA was characterized further in detail. The purification of myrosinase after the chromatography step was 169-fold, with 37% recovery of the total activity in the crude extract. 

The purification efficiency of the presented method was higher than the efficiency of routine purification protocols, which use the ammonium sulfate precipitation combined with standard chromatographic techniques, such as ion-exchange chromatography and gel filtration, and which make it possible to obtain the final myrosinase(s) with the purification factor up to 34-fold [18,20]. Several studies have shown that the replacement of standard chromatographic steps or their combination with lectin affinity chromatography, using concanavaline A as the ligand, leads to a pronounced increase in the purification efficiency of the methods, providing a purification level of the final myrosinases ranging from 40- [38] to 1318-fold [39]. Our method, although containing only one chromatographic step following isoelectric precipitation by ammonium sulfate, achieved a sufficient purification factor of 169-fold. 

The purity and homogeneity of purified myrosinase was evaluated by polyacrylamide gel electrophoresis in the presence of SDS under non-reducing conditions (Figure 3B). The electrophoretic analysis revealed two protein bands comigrating with myrosinase activity conspicuous on zymography gel, one major and one minor, with molecular weights of 113.6 and 122.1 kDa, respectively (relative molecular weights of bands were calculated from the linear relationship between the relative mobility of the protein standards and logarithm of their molecular masses). It is known that enzyme in the native form exists as a homodimer and is substantially glycosylated [40]. The difference in the molecular weight of both bands likely reflects a variation in the level of glycosylation. Another band, with no corresponding myrosinase activity, appearing at the molecular weight of 58.3 kDa, may represent the size of the enzyme monomer or one of the regulatory proteins, such as myrosinase-binding proteins and myrosinase-associated proteins. Several such regulatory proteins, with sizes from 30 to 110 kDa, have been identified in rapeseed (*Brassica napus*) [41,42,43], and bind to myrosinases and specify how they could cleave glucosinolates.

The data show that myrosinase from *L. sativum* can be obtained in convenient yield and with electrophoretic purity by a simple purification procedure based on the one-step affinity chromatography using SFN as ligand.

### 3.2. Identification of Myrosinase by Mass Spectrometry Analysis

Identification of myrosinase was performed by two methods, immunochemically using a polyclonal antibody against myrosinase (TGG1) from *A. thaliana* (anti-TGG1; obtained from commercial source), and the MALDI-Tof-Tof technique. For immunochemical detection, the protein bands, obtained after electrophoresis of the purified enzyme on the polyacrylamide gel, were transferred by Western blot to a nitrocellulose membrane and then detected with anti-TGG1 antibody. However, the anti-TGG1 antibody interacted with recombinant myrosinase TGG1 [44], the immunodetection of purified myrosinase from *L. sativum* with same antibody failed (not shown). This observation indicates that purified myrosinase from *L. sativum* and TGG1 myrosinase from *A. thaliana* do not share a high degree of sequence homology.

Major protein band on polyacrylamide gel co-migrating with myrosinase activity was trypsinized, and the resulting peptide fragments used for identification by MALDI-Tof mass spectrometry. However, the tryptic peptide digest profile did not provide a positive identification for myrosinase or any other homologous proteins in the NCBI and/or SwissProt databases. To obtain some information about the primary structure of purified myrosinase, three major tryptic peptides were further subjected to secondary fragmentation and analyzed by tandem mass spectrometry (not shown). The following peptide sequences were identified (NHNADVAVDFYHR, HWITFNEPWVFSR, IGIAHSPAWFEPEDVEGGQNTVDR) and used for multiple sequence alignment with sequences of known plant myrosinases, QE- and EE-type (atypical myrosinases), and β-D-glucosidases. 

The results emerging from comparative analysis show that the identified peptide sequences from *L. sativum* myrosinase are congruous to the regions responsible for the binding of glycoside substrates and catalytic functions of both mentioned classes of enzymes (Figure 4). The amino acid sequences NHNADVAVDFYHR and HWITFNEPWVFSR are entirely identical to the corresponding regions of β-D-glucosidases and exhibit 80% and 100%, respectively, sequence homology with atypical myrosinases (EE-type myrosinases), but only 40% and 57% sequence homology with QE-type myrosinases. In particular, the presence of the conserved sequence motif TFNEP, which is part of the second peptide sequence, is key for the catalytic mechanism of hydrolysis of the β-glycosidic bond in the GH1 glycoside hydrolase family [45]. The third identified peptide sequence (IGIAHSPAWFEPEDVEGGQNTVDR) is more variable, and its sequence homology with β-glucosidases ranges from 60% (for BGL18_ARATH) up to 100% (for XP_006406276). Sequence homology with atypical myrosinases achieved 65% for PYK10 or 71% for PEN2, and less than 43% (for TGG1) sequence homology with QE-type myrosinases.

Mass spectrometry analysis of the trypsinized fragments of the purified enzyme from *L. sativum* revealed the conserved sequence motifs characteristic for the glycoside hydrolase family 1 (GH1) members involved in the substrate binding site and the cleavage of the glycosidic bond.

### 3.3. Enzymatic Properties of Purified Myrosinase

Kinetic parameters of hydrolysis of native glucosinolates by myrosinase reaction were determined by nonlinear regression of enzyme kinetic data, which were fitted to the Michaelis–Menten equation. The obtained results, summarized in Table 2, showed that the enzyme is able to cleave all tested substrates (sinigrin, glucoraphanin and sinalbin), but with slightly different affinity and velocity of hydrolysis for each of them.

The greatest affinity of myrosinase was observed for sinigrin with a K_M_ of (0.57+/−0.24) mM. For sinalbin and glucoraphanin, the affinity was reduced about 10% and 50%, respectively. The maximal rate (V_max_) of glucosinolate hydrolysis was reached towards the aliphatic substrates, glucoraphanin and sinigrin, and lower towards aromatic sinalbin. The catalytic efficiency (k_cat_/K_M_) of enzyme for the hydrolysis of thioglycosidic linkage in glucosinolates was the highest against sinigrin and decreased by 40% and by 47% against sinalbin and glucoraphanin, respectively.

Due to the ability of myrosinases to hydrolyze various glucosinolates, several synthetic O-glycoside analogs with structural similarities to glucosinolates were tested as substrates (Table 3).

Previous studies have shown that L-ascorbic acid has no effect on the myrosinase-catalyzed cleavage of nitrophenyl-O-glycosides [46,47]; therefore, these measurements were performed in the absence of L-ascorbic acid. Among O-glycosides, the uppermost level of specific enzyme activity (0.017 ± 0.001 µmol/min/mg) was measured against pNPG; nevertheless, this value-specific activity was more than 7 times lower in comparison with the one obtained for natural thioglycoside, sinigrin, under the same reaction conditions. Other O-glycoside analogs differing in the type of monosaccharide residue (with D-galactopyranose or D-mannopyranose in molecule), the configuration of the glycosidic linkage (α- and β-anomer), or the location of a nitro group on the benzene ring were not substrates suitable for isolated myrosinase and were not hydrolysed at all (Table 3). 

Our observation is consistent with that of Durham and Poulton [48], who also showed that myrosinase isolated from *L. sativum* seedlings hydrolyzes pNPG at 18% of the rate observed with sinigrin. The difference in the rate of hydrolysis of sinigrin and pNPG could be attributed to the ability of purified myrosinase to bind both substrates. pNPG substrate does not have the sulphate group, which is the important for the recognition and binding of the aglycone moiety of glucosinolates by myrosinases, as shown by the crystallographic analysis of the structure of *S. alba* myrosinase [31,40]. 

This hypothesis was experimentally corroborated by Anderson et al. [49], who showed that the affinity of myrosinase isoenzymes (TGG1, TGG4 and TGG5) in *A. thaliana*) for pNGP was 756-, 114- and 146-times, respectively, lower than that of sinigrin, and that to achieve their complete saturation with pNGP is very difficult. 

The results described above show that myrosinase from the seeds of *L. sativum* is able to split both types of β-glycosides, S- and O-, at different levels, but the presence of the glucose moiety, and the β-configuration of the glycosidic linkage, formed between the C-1 atom of D-glucose and aglycone, in the substrate molecule are favorable structural characteristics for the successful cleavage of the glycosidic bond.

The temperature dependence of the myrosinase activity showed that enzyme is active in a relative wide range of temperatures (from 20 up to 80 °C), with an apparent optimum at 50 °C (Figure 5A).

The optimal pH range for myrosinase activity was in the moderately acidic region ranging from 5.2 to 6.2, the maximum activity was achieved at pH of 6.0 (Figure 5B). The analysis of thermostability showed that the level of the remaining activity of myrosinase after incubating the enzyme at various temperatures in the range from 4 to 50 °C for 60 min is similar with negligible differences, but the activity dropped almost to the background level when enzyme was incubated at 70 °C (Figure 5C).

The request of L-ascorbic acid for the stimulation of myrosinase activity was tested in the concentration range from 0.1 μM to 5 mM. The effect of L-ascorbate on the myrosinase activity had a typical biphasic character (Figure 5D). The stimulation of enzyme activity was observed up to 30 μM, with its optimum in the presence of 30 μM L-ascorbate (with the increase of specific activity more than 12-fold in comparison to the control measurement). However, further increase of L-ascorbate concentration led to the inhibition and, at 2 mM level, to the complete loss of enzyme activity. The purified enzyme was found to be stable, and it could be stored for more than one month at 4 °C without any pronounced change in specific activity (Figure 5E). However, freezing of the enzyme (at −20 or −80 °C) was accompanied by the loss of enzyme activity. The loss of enzyme activity of *L. sativum* myrosinase upon the storage at −20 °C was also observed by [50].

The stability of myrosinase is a key application parameter; therefore, the thermal properties of myrosinases prepared from leaves or seeds of various *Brassica* vegetables, such as broccoli [51,52], red cabbage [53,54], green cabbage [55], white cabbage [54], chinese flowering cabbage [56], white mustard [57], and black, brown and yellow mustard [58], have been studied. These studies showed that although their resistance against thermal inactivation is different, almost all enzymes, including enzyme purified from *L. sativum* (this work), retain their stability in temperatures ranging from 10 to 40 °C, and some of them, such as red cabbage and white mustard, at temperatures up to 60 °C [54,57]. It is assumed that the thermal stability of myrosinases is a result of the presence of numerous salt bridges, hydrogen bonds, and disulphide bridges, formed between amino acid residues situated at the surface of enzyme [31]. Moreover, the experiments with the recombinant broccoli myrosinases, produced in *Saccharomyces cerevisiae* and *Escherichia coli* and differing in the level of glycosylation, revealed the diverse effects of temperature on enzyme activities of both myrosinases [59]. This observation suggests that glycosylation may play some role in thermal stability of myrosinase too.

In the production of enzymes for food and medical purposes, we often encounter distrust of products prepared by recombinant technologies. However, this route makes it possible to prepare larger amounts of pure enzyme. On the contrary, in the preparation of enzymes from plant sources, in addition to time-consuming purification, low enzyme yield seems to be another disadvantage. Therefore, both approaches are currently relevant. In the last two decades, heterologous expressions of the myrosinase (TGG1-TGG5) genes from *Arabidopsis thaliana* [19,44,49,60,61] and from *Carica papaya* [46] have been successfully performed in yeast (*Pichia pastoris* and *Yarrowia lipolytica*). Broccoli myrosinase genes were expressed in *E. coli* and *S. cerevisiae* [59]. In the present work, we propose a relatively simple method for isolating myrosinase from *L. sativum*, providing electrophoretically homogeneous preparation. This preparation could be proteomically analyzed by MALDI-Tof-Tof technology. The obtained protein alignments make it possible to design a method for obtaining the gene of this myrosinase for possible recombinant preparation.

## 4. Conclusions

The main objective of this study was to develop an effective method for the purification of myrosinase from *L. sativum*. An inventive purification procedure, involving the isoelectric precipitation with AS and specific SFN affinity-based chromatography, for the preparation of *L. sativum* myrosinase with electrophoretic purity (with the final purification of 169-fold and 37% recovery) and its optimization was presented. The close relationship of the purified enzyme with the members of β-glucohydrolase family 1 was proven by the data from MALDI-tof-tof mass spectrometry analysis and kinetic measurements with native and synthetic substrates. This analysis revealed that purified enzyme preferentially cleaves the S-glycosidic bond in glucosinolates and its activity is dependent on the presence of L-ascorbate. Key factors affecting the hydrolytic activity of the enzyme, such as optimum pH and temperature, thermal inactivation, and long-term storage stability, were characterized too. This purification approach makes it possible to obtain a purified enzyme without the use of recombinant techniques from natural sources, albeit in smaller yield.

## Data Availability

Additional data are available from the authors.

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
