# Peer review of "Sulphoraphane Affinity-Based Chromatography for the Purification of Myrosinase from Lepidium sativum Seeds"

_biomolecules, 2022, doi:10.3390/biom12030406_

Round 1
Reviewer 1 Report
In this paper, the purification of myrosinase was realized by SFN ligand. I would like to request addressing the following issues.
1.L37:Does "cutting enzyme properties" refer to myrosinase or some proteins such as ESP that are involved in glycoside rearrangement in plants? Some studies have shown that glycoside rearrangement is a chemical reaction and is not affected by the hydrolysis of myrosinase in the first step, so whether there is corresponding research support to change the expression.
2. Is the purification of myrosinase by SFN ligand a first? If not, appropriate references should be added. At the same time, only the purification efficiency of this study was discussed in this paper, which can be appropriately added to the discussion and compared with other current studies on the purification efficiency of myrosinase.
3. The temperature interval of the optimal temperature is 20℃, and if the interval is too large, the error of the optimal temperature is large. Maybe you can set the interval at 5℃.
4. The stability of this enzyme can be suitably compared with other myrosinases.
5. To further improve the quality of this manuscript, some new papers, such as JournalofAgricultural and Food Chemistry, 2021, 69: 5363-5371, should be Should be cited and discussed.
Author Response
Thank you for your valuable recommendations.

Reviewer 2 Report
The main goal of the reviewed work was to develop an effective treatment method
myrosinase from L. sativum.
The introduction to the publication was well written on the basis of 22 references.
The experiment was properly planned. The research methodology is well described. The results are presented in three tables and five figures. The obtained research results were thoroughly discussed. The aim of the research has been achieved. The conclusions are correct.
The overall assessment of the manuscript is positive. However, before publishing, authors should correct the manuscript:
1) What error are the results presented in Figure 1.
2) Provide the value of +/- SD for the measured parameters listed in Table 1.
Author Response
Reply: Thanks for your suggestions. We have calculated the standard deviations and have added them to graph (Figure 1) and Table 1.
Reviewer 3 Report
The manuscript describes the purification and characterization of myrosinase from Lepidium sativum seeds. In general d ethe work is well done and will be of interest to those working in health-promoting benefits of plant isothiocyanates. However, there are some points to be considered.
1.- The molar extinction coefficient of pNPP is 18000 M-1cm-1, why the authors used 9425 M-1cm-1?, please correct the values in Table 3.
2.- In the same context, which was the enzyme activity after incubating 4 h at 37°C?, because maybe the enzyme lost its activity only by the conditions, and the values showed in table 3 are underestimated and not related with the different substrates. Additionally, which was the specific activity for sinigrin measured at normal conditions (Table 2)?
3.- In figure 5, the assays were performed in the presence of 1 mM L‐ascorbic acid, but according to subsection D, this concentration affects enzyme activity, again, the values reported in the other subsections could be wrong.
4.- In material and methods is stated that GD method was used to determine the kinetic parameters but in Table 2 is stated that these were determined by GOP method. Which is correct?
5.- Line 388 should be deleted, is an incomplete phrase.
6.- Please improve the quality of figure 4, it is difficult to read.
7.- Please discuss about the quantity of enzyme obtained at the end of the purification protocol (0.44mg), if 20 µg of enzyme were used in each measure, several purifications need to be performed. What happened if 500 g of seeds are used instead 10 g, the purification protocol still works or which was the reason to use 10 g?
Author Response
Thank you for your valuable recommendations

Round 2
Reviewer 1 Report
The manuscript has been carefully revised and can be accepted in present form.
Reviewer 3 Report
The manuscript was corrected according the suggestions. Therefore, I recommend ist publication.